# Analyzing and Boosting the Power of Fine-Grained Visual Recognition for Multi-modal Large Language Models

**Hulingxiao He**[1], **Geng Li**[1], **Zijun Geng**[1], **Jinglin Xu**[2], **Yuxin Peng**[1]*

[1]Wangxuan Institute of Computer Technology, Peking University

[2]School of Intelligence Science and Technology, University of Science and Technology Beijing

`{hehulingxiao,ligeng}@stu.pku.edu.cn, gengzijun2024@163.com`
`xujinglinlove@gmail.com, pengyuxin@pku.edu.cn`

## Abstract

Multi-modal large language models (MLLMs) have shown remarkable abilities in various visual understanding tasks. However, MLLMs still struggle with fine-grained visual recognition (FGVR), which aims to identify subordinate-level categories from images. This can negatively impact more advanced capabilities of MLLMs, such as object-centric visual question answering and reasoning. In our study, we revisit three quintessential capabilities of MLLMs for FGVR, including object information extraction, category knowledge reserve, object-category alignment, and position of the root cause as a misalignment problem. To address this issue, we present **Finedefics**, an MLLM that enhances the model's FGVR capability by incorporating informative attribute descriptions of objects into the training phase. We employ contrastive learning on object-attribute pairs and attribute-category pairs simultaneously and use examples from similar but incorrect categories as hard negatives, naturally bringing representations of visual objects and category names closer. Extensive evaluations across multiple popular FGVR datasets demonstrate that Finedefics outperforms existing MLLMs of comparable parameter sizes, showcasing its remarkable efficacy. The code is available at `https://github.com/PKU-ICST-MIPL/Finedefics_ICLR2025`.

## 1 Introduction

Multi-modal Large Language Models (MLLMs) (Bai et al., 2023; Chen et al., 2023; Zhang et al., 2023b; Zhu et al., 2023; Dong et al., 2024; Liu et al., 2024b;a; Laurençon et al., 2024a;b) have achieved remarkable advancements in understanding visual data, showcasing potential in advancing general artificial intelligence. These models enable users to interact with images as inputs, fostering seamless communication grounded in visual information. The impressive capabilities allow MLLMs to excel in various vision tasks while adeptly handling complex content comprehension and generation. However, despite their versatility and linguistic proficiency, MLLMs still face challenges in a fundamental task of machine vision: fine-grained visual recognition (FGVR) (Zhang et al., 2024b; Geigle et al., 2024), which aims at identifying subordinate-level categories, such as specific species of animals or plants (Wei et al., 2021). Poor FGVR performance of MLLMs hinders them from performing more advanced tasks like object-centric visual question answering and reasoning (Zhang et al., 2024b). For example, in smart agriculture, poor FGVR performance of pests may lead to incorrect treatment strategies and large-scale reduction in food production.

Early works have investigated the phenomenon (Zhang et al., 2024b) and attempted to improve the FGVR performance of MLLMs by integrating open-set classification data into pre-training or fine-tuning stage (Geigle et al., 2024; Zhang et al., 2024b). However, fine-tuning solely on the classification task harms the general capability of the instruction following, while purely integrating classification-focused data into the instruction tuning data brings limited improvement (Geigle et al., 2024), making their direct utilization impractical. To understand why MLLMs underperform in

---

*Corresponding author.

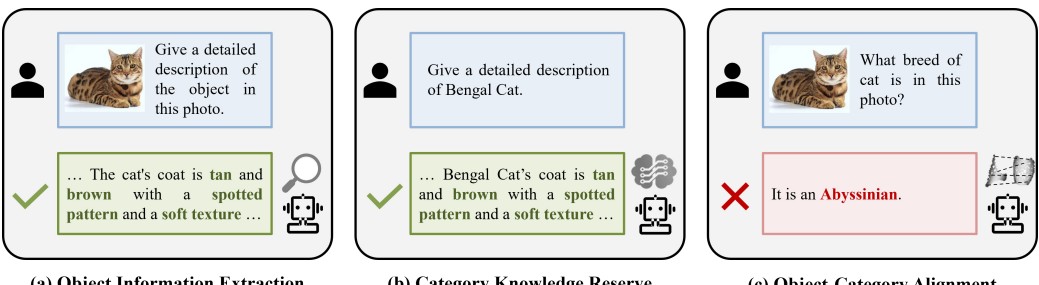

**(a) Object Information Extraction**   **(b) Category Knowledge Reserve**   **(c) Object-Category Alignment**

Figure 1: Three quintessential capabilities of MLLMs for fine-grained visual recognition. Current MLLMs possess acceptable capabilities in image information extraction and category knowledge reserve but struggle with aligning objects to their corresponding subordinate-level categories.

FGVR, we revisit and evaluate three quintessential capabilities of MLLMs as shown in Figure 1: (a) **Object information extraction**. It is essential to accurately and fully extract the necessary information for distinguishing objects. (b) **Category knowledge reserve**. MLLMs should reserve sufficient knowledge of subordinate-level categories. (c) **Object-category alignment**. With rich visual information extracted and sufficient category knowledge reserved, visual objects and category names should be aligned in the representation space to enhance classification performance.

We analyze the representation space of MLLMs and their corresponding visual language models (VLMs) like CLIP (Radford et al., 2021), revealing that: (1) *Object information lost exists between VLMs and MLLMs but is not the bottleneck.* During the propagation of object features output from the vision encoder in modality connector and language model layers, the necessary visual information for distinguishing objects is almost preserved. Our observation is consistent with object hallucination in MLLMs (Zhou et al., 2023), which finds that the modality connector tends to decrease the representation discriminability, while LLM has no significant impact. (2) *Category knowledge is relatively sufficient, but category names cannot fully capture the semantics.* Existing LLMs utilized in MLLMs can output detailed and distinguishing descriptions about subordinate-level categories, but category names are not discriminative in the representation space of LLMs. (3) *Misalignment between the visual object and category name leads to underperformance.* Despite a slight reduction in both object and category representation discriminability, the current learned modality connector is insufficient for effectively matching visual object representations to subordinate-level category names, as most VLMs do.

Motivated by the aforementioned analysis, we propose **Finedefics**, an MLLM designed to enhance the model's ability to identify subordinate-level visual object categories. Our framework builds upon Idefics2 (Laurençon et al., 2024b) and is specifically tailored to boost the power of FGVR. To facilitate alignment between visual objects and category names, descriptions summarized from information visual attributes are utilized as the intermediate point to bind them in the representation space of LLMs. Concretely, we separately feed token sequences of visual objects, attribute descriptions, and category names into MLLMs to obtain global representations from the last layer of LLMs respectively. Contrastive learning is performed on global representations of object-attribute pairs and attribute-category pairs simultaneously, with additional hard negatives from similar but incorrect categories, bringing representations of visual objects and category names closer. Subsequently, being trained solely on both open-set and closed-set FGVR data through instruction tuning, Finedefics demonstrates exceptional FGVR performance gains. Benefiting from attribute augmented alignment, Finedefics outperforms established counterparts across six popular FGVR datasets and notably surpassing Idefics2 (Laurençon et al., 2024b) and Qwen-VL-Chat (Bai et al., 2023) by an average of +10.89% and +9.43%, respectively.

In summary, our contributions are as follows: (i) We revisit the quintessential capabilities of MLLMs for FGVR and investigate the root cause of underperformance in FGVR: misalignment between visual objects and category names. (ii) We propose Finedefics for enhancing the model's FGVR accuracy, which uses informative attribute descriptions to effectively align visual objects and category names in the representation space of LLMs. (iii) With extensive experiments on six popular FGVR datasets, we demonstrate the superiority of Finedefics.

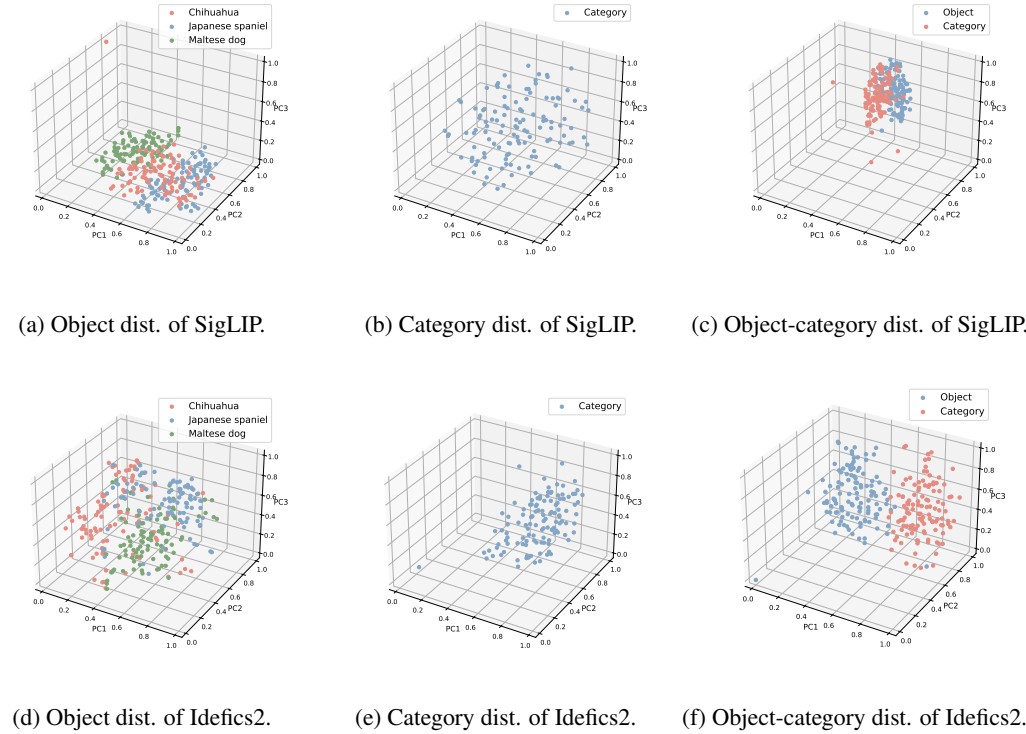

(a) Object dist. of SigLIP.   (b) Category dist. of SigLIP.   (c) Object-category dist. of SigLIP.

(d) Object dist. of Idefics2.   (e) Category dist. of Idefics2.   (f) Object-category dist. of Idefics2.

Figure 2: Object/Category/Object-category representation visualization of SigLIP and Idefics2.

## 2 WHY DO MULTI-MODAL LARGE LANGUAGE MODELS UNDERPERFORM IN FINE-GRAINED VISUAL RECOGNITION?

This section scrutinizes the root causes of underperformance in FGVR via comprehensive empirical analyses on three quintessential capabilities of MLLMs: *object information extraction*, *category knowledge reserve*, and *object-category alignment* by comparing the representation space with corresponding VLMs (Idefics2 (Laurençon et al., 2024a) and SigLIP (Zhai et al.) in our experiments).

**Notations.** Assuming an image $I_i$ containing an object $O_i$ is processed by the vision encoder $\mathbf{V}_\alpha$ and learnable modality connector $\mathbf{F}_\beta$ to be transformed into a visual object token sequence of length $m$: $S_o^i = [o_1^i, o_2^i, \ldots, o_m^i]$. Input category name in textual modality $C_i$ is passed through an embedding layer $E_\phi$ of the LLM to obtain the category embedding sequence of length $n$: $S_c^i = [c_1^i, c_2^i, \ldots, c_n^i]$. Subsequently, the object embedding sequence $S_o^i$ and category embedding sequence $S_c^i$ are individually passed through the LLM layers $\mathbf{L}_\theta$ to obtain the output from the last layer:

$$H_o^i = \mathbf{L}_\theta(S_o^i), \tag{1a}$$

$$H_c^i = \mathbf{L}_\theta(S_c^i), \tag{1b}$$

where $H_o^i = [\hat{o}_1^i, \hat{o}_2^i, \ldots, \hat{o}_m^i]$, and $H_c^i(l) = [\hat{c}_1^i, \hat{c}_2^i, \ldots, \hat{c}_n^i]$. Afterward, we select two ways to represent the global semantics of output sequence following (Zhang et al., 2024b): 1) last token embedding $\hat{o}_m^i$, $\hat{c}_n^i$, and (b) average of the token embedding sequence $\bar{o}^i = (\sum_{k=1}^m \hat{o}_k^i)/m$, $\bar{c}^i = (\sum_{k=1}^n \hat{c}_k^i)/n$. For VLMs, the projected [CLS] embedding outputs from last layer of vision encoder $\mathbf{V}_\alpha$ and text encoder $\mathbf{T}_\gamma$ are taken to represent the global semantics of $O_i$ and $C_i$, denoted as $\hat{o}_{\text{CLS}}^i$ and $\hat{c}_{\text{CLS}}^i$, respectively.

### 2.1 OBJECT INFORMATION EXTRACTION

In the task of FGVR, a model that excels at object information extraction is required to have discriminative representations, i.e., large inter-class distance and small intra-class variance. To com-

Table 1: Feature probing on Idefics2 and SigLIP with features of objects and category descriptions.

(a) Object features.

| Model | Feature Type | Acc. |
|---|---|---|
| Idefics2 | Last | 94.99 |
| | Avg. | 90.24 |
| SigLIP | CLS | 95.28 |
| | Avg. | 94.44 |

(b) Category description features.

| Model | Feature Type | Acc. |
|---|---|---|
| Idefics2 | Last | 92.51 |
| | Avg. | 90.41 |
| SigLIP | CLS | 84.70 |
| | Avg. | 87.78 |

pare the object representation space of SigLIP and Idefics2, we select three subordinate-level categories `["Chihuahua","Japanese spaniel","Maltese dog"]` from Stanford Dog-120 (Krause et al., 2013), and randomly sample 100 examples per category for t-SNE (Van der Maaten & Hinton, 2008) visualization. As shown in Figure 2a, since the vision encoder $\mathbf{V}_\alpha$ is normally frozen throughout training (Liu et al., 2024b;a), the output object token sequence $S_o^i$ preserves discriminative information for classification. We then hypothesize that the information is lost after propagating through the modality connector $\mathbf{F}_\beta$ and LLM layers $\mathbf{L}_\theta$. However, various objects belonging to the same subordinate-level categories can still cluster together and distance from each other, as illustrated in Figure 2d. To quantitatively compare the representation discriminability, we use feature probing experiments (Zhang et al., 2024b) to test the hypothesis. Concretely, on top of the last token embedding $\hat{o}_m^i$ or average of the token embedding sequence $\bar{o}^i$, we train a linear classifier on the training set of Oxford-IIIT Pet-37 (Parkhi et al., 2012) and evaluate on the test set. In Table 1a, we observe that although information is lost, the impact on the performance is limited.

## 2.2 CATEGORY KNOWLEDGE RESERVE

Trained on enormous internet-scale corpora, LLMs are known for encoding the expert knowledge for general categories in their weights, but we ask ourselves, is the expert knowledge quintessential for FGVR already contained in MLLMs? We hypothesize that MLLMs' underperformance in FGVR tasks stems from the inadequate knowledge of subordinate-level categories. To test the hypothesis, we investigate whether LLMs utilized in MLLMs can distinguish different categories by generating discriminative descriptions. Specifically, we probe the knowledge in Idefics2 via using the prompt `["Give a brief description of distinguishing features of {CLASS NAME}"]`. For each subordinate-level category in Oxford-IIIT Pet-37 (Parkhi et al., 2012), we set the number of return descriptions to 200 and then equally divided them into the train set and test set. The class names in returned descriptions are replaced with demonstrative pronouns to avoid leakage of classification labels. Similarly, we conduct linear probing experiments on top of $\hat{c}_n^i$ and $\bar{c}^i$. As shown in Table 1b, Idefics2 exhibits better classification performance than the text encoder of SigLIP, demonstrating its superiority in reserving category knowledge. Despite the rich semantics of the generated category description, the category names have lower discriminability in the representation space of Idefics2 than the text model of SigLIP, illustrated in Figure 2b and 2e.

## 2.3 OBJECT-CATEGORY ALIGNMENT

Since our empirical study shows that Idefics2 has an acceptable capability of object information extraction and adequate knowledge of subordinate-level categories, we hypothesize that the misalignment between the visual object and category name is the root cause. We randomly sample 120 object-category pairs from Stanford Dog-120 (Krause et al., 2013), and visualize the distributions of the last token embedding of the object $\hat{o}_m^i$ and the category name $\hat{c}_m^i$ in the same representation space. As shown in Figures 2c and 2f, object and category representations have significant semantic gaps. Since category names may not fully represent the semantics of the visual data (Lyu et al., 2024), the object cannot match the ground-truth category in the representation space and thus fails to decode into the correct category name.

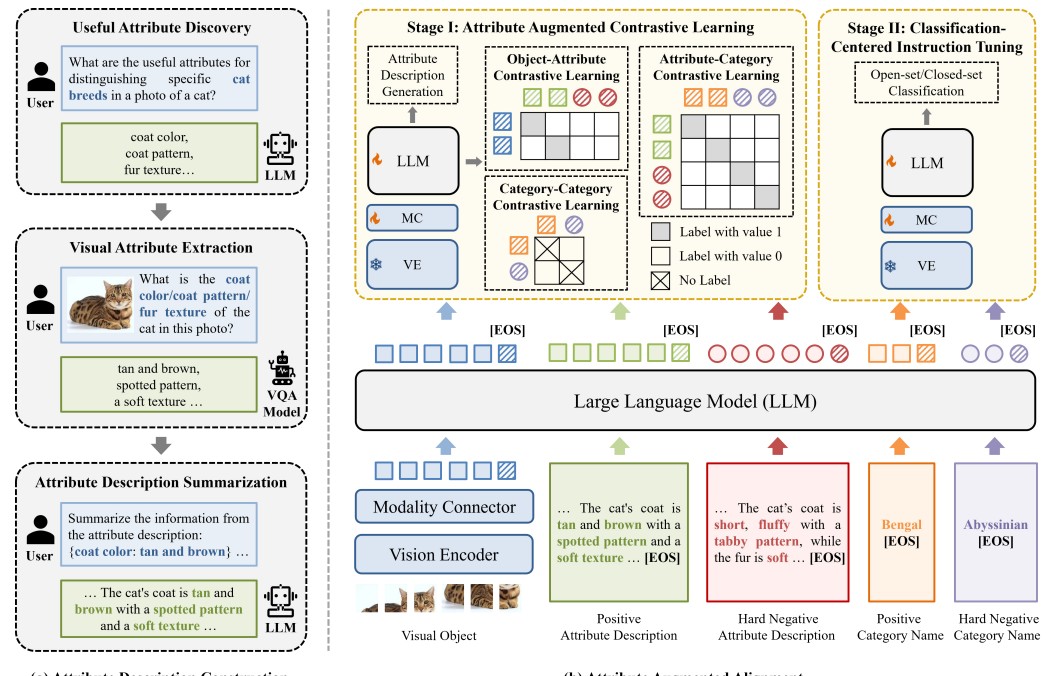

Figure 3: An illustration of framework to build Finedefics. (a) Attribute Description Construction, which aims to obtain informative attribute descriptions of objects. (b) Attribute Augmented Alignment, which aims to use constructed attribute descriptions to bind visual objects and category names, thus enhancing the model's FGVR capability via a two-stage training paradigm.

## 3 METHOD

After thoroughly investigating the root cause of the underperformance in FGVR, this section formally introduces Finedefics, which enhances the model's FGVR performance by better aligning visual objects and category names. The framework to build Finedefics is illustrated in Figure 3, composed of two key components: (1) Attribute Description Construction for extracting useful attribute information that can distinguish different categories. (2) Attribute Augmented Alignment dedicated to using constructed attribute descriptions as the intermediate point to bind visual objects and category names in the representation space of LLMs, thus boosting the subsequent Classification-Centered Instruction Tuning.

### 3.1 ATTRIBUTE DESCRIPTION CONSTRUCTION

Although it has been demonstrated in (Liu et al., 2022; El Banani et al., 2023) that language is a powerful tool for capturing semantic relationships, dependency on category names exclusively to align with extracted visual embeddings is unreliable. As discussed in Section 2.2, leveraging adequate knowledge of subordinate-level categories, LLMs can describe distinguishing features that better capture category semantics than category names. Inspired by (Liu et al., 2024c) that exploits a cascade of foundation models to translate useful visual information from visual to textual modality, we propose constructing sample-wise attribute descriptions for each FGVR training set.

Specifically, the construction comprises three steps: 1) Useful Attribute Discovery by LLMs, such as GPT-4 (Achiam et al., 2023) and LLaMA (Touvron et al., 2023). These attribute names are employed as keys to instruct Visual Question Answering (VQA) models (such as BLIP-2 (Li et al., 2023) and LLaVA (Liu et al., 2024b)) for extracting useful attribute values. 2) Visual Attribute Extraction by VQA models. These attribute values enrich the information for distinguishing subordinate-level categories. 3) Attribute Description Summarization by LLMs. These descriptions help alleviate the gaps between visual objects and category names in the training phase of Finedefics.

**Userful Attribute Discovery.** Given the super-category of each FGVR dataset, such as `aircraft` for FGVC-Aircraft (Maji et al., 2013), we first identify a useful set of attributes that set apart the subordinate-level categories. For example, the `wing shape` attribute can help distinguish various aircraft models. To discover such key visual cues, we tap into the expert knowledge of LLMs, which is otherwise only restricted to experts. Specifically, we ask LLMs: `["Your task is to tell me what are the useful attributes for distinguishing {SUPERCLASS} {CLASSUNIT} in a photo of a {SUPERCLASS}"]`. Formally, LLM takes a super-category $C_{\text{sup}}$ as input and outputs a list of useful attributes:

$$N^{C_{\text{sup}}} = \mathbf{L}_\theta \left( P^{\text{dis}}(C_{\text{sup}}) \right), \tag{2}$$

where $N^{C_{\text{sup}}} = \{N_1^{C_{\text{sup}}}, \ldots, N_s^{C_{\text{sup}}}\}$ are the generated attribute keys for the category $C_{\text{sup}}$, $\mathbf{L}_\theta$ are LLM layers, and $P^{\text{dis}}$ is the How-to LLM-prompt in (Liu et al., 2024c).

**Visual Attribute Extraction.** With the discovered attribute names $N^{C_{\text{sup}}}$, we leverage VQA models that excel at identifying general visual attributes (e.g., shape, color) of objects to extract each attribute value per sample. For example, if an attribute is `wing shape`, VQA models are prompted to give a brief description of the wing shape, which is a much easier task than recognizing many subordinate-level categories. Following (Liu et al., 2024c), we add a general attribute name $N_0^{C_{\text{sup}}} = $ `["General description of the image"]` and its prompt $P_0^{\text{ext}} = $ `["Questions: Describe this image in details. Answer:"]`. Formally, VQA model takes as input an image $I_i$, its super-category $C_{\text{sup}}^i$ and the attribute names $N^{C_{\text{sup}}^i}$, the output visual attributes are given as:

$$V_i = \mathbf{Q}_\epsilon \left( I_i, C_{\text{sup}}^i, P_{\text{ext}}(N^{C_{\text{sup}}^i}) \right), \tag{3}$$

where $V_i = \{V_1^i, \ldots, V_v^i\}$ denotes the extracted set of visual attributes for image $I_i$, $\mathbf{Q}_\epsilon$ is the VQA model, and $P_{\text{ext}}$ is the Identify VQA-prompt in (Liu et al., 2024c).

**Attribute Description Construction.** After obtaining the structured set of attribute key-value pairs, we further ask LLMs: `["Summarize the information you get about the {SUPERCLASS} from the general description and attribute description with five sentences."]`. The summarized attribute description contains richer semantics of subordinate-level categories, making it much easier for LLM to understand. Formally, given the set of attribute names $N^{C_{\text{sup}}^i}$ and attribute values $V_i$, LLM outputs a summarized attribute description for image $I_i$:

$$A_i = \mathbf{L}_\theta \left( P_{\text{con}}(N^{C_{\text{sup}}^i}, V_i) \right), \tag{4}$$

where $A_i$ is the attribute description constructed for image $I_i$, and $P_{\text{con}}$ is the revised Reason LLM-prompt in (Liu et al., 2024c) for summarization task only. Expanding upon our newly built attribute descriptions, we transfer traditional (object, category) pairs in FGVR datasets to (object, attribute, category) triples. Without specification, the category refers to the subordinate-level category instead of the super-category in subsequent sections.

## 3.2 Attribute Augmented Alignment

With the constructed informative attribute descriptions, we introduce a new training paradigm named Attribute Augmented Alignment to build our Finedefics. It comprises two stages: (I) Attribute Augmented Contrastive Learning for aligning visual objects and category names in the representation space of LLMs. (II) Classification-Centered Instruction Tuning for enhancing the model's ability to follow the FGVR task instruction.

**Stage I: Attribute Augmented Contrastive Learning.** For each object-attribute-category triple $(O_i, A_i, C_i)$, we utilize the vision encoder $\mathbf{V}_\alpha$ and the learnable modality connector $\mathbf{F}_\beta$ to transfer $O_i$ into an object embedding sequence of length $S_o^i = [o_1^i, o_2^i, \ldots, o_m^i]$ with length $m$. To better capture the global representations, we follow (Jiang et al., 2024) to pass an `[EOS]` token through an embedding layer $\mathbf{E}_\phi$ of LLM to obtain the vector representation and append it to the visual embedding sequence $S_o^i$. Therefore, we obtain the newly built object embedding sequence $\tilde{S}_o^i = [o_1^i, o_2^i, \ldots, o_m^i, o_{\text{EOS}}^i]$. Similarity, we obtain the attribute embedding sequence $\tilde{S}_a^i = $

$[a_1^i, a_2^i, ..., a_p^i, a_{\text{EOS}}^i]$ with length $(p+1)$, and category embedding sequence $\tilde{S}_c^i = [c_1^i, c_2^i, ..., c_n^i, c_{\text{EOS}}^i]$ with length $(n+1)$. Then, $\tilde{S}_o^i, \tilde{S}_a^i, \tilde{S}_c^i$ are individually fed into LLM layers $\mathbf{L}_\theta$, and the embeddings of the last predicted token $\hat{o}_{\text{EOS}}^i, \hat{a}_{\text{EOS}}^i, \hat{c}_{\text{EOS}}^i$ are utilized as the global representations of $O_i, A_i, C_i$, respectively. Without specified, we use $\hat{o}^i = \hat{o}_{\text{EOS}}^i, \hat{a}^i = \hat{a}_{\text{EOS}}^i, \hat{c}^i = \hat{c}_{\text{EOS}}^i$ for simplicity.

To improve the effectiveness of contrastive learning, we then mine difficult incorrect category names for each example object $O_i$ used in the FGVR dataset. To do this, we use a CLIP model (Radford et al., 2021) for mining hard negative samples: for every example image, we select three images along with their attribute descriptions from the three most similar but incorrect categories. Attribute descriptions and category names from these hard negative samples are subsequently treated as additional negatives. Thus, the formulation of Object-Attribute Contrastive (OAC) loss with the inclusion of hard negatives can be described as follows:

$$\mathcal{L}_{OA}^{hn} = \sum_{(\hat{o}^i, \hat{a}^i, \hat{c}^i) \in \mathcal{B}} -\log \frac{\exp^{Sim(\hat{o}^i, \hat{a}^i)}}{\sum\limits_{\hat{a}^j \in \mathcal{B}} \exp^{Sim(\hat{o}^i, \hat{a}^j)} + \sum\limits_{\hat{a}^w \in \mathcal{A}_{hn}^i} \exp^{Sim(\hat{o}^i, \hat{a}^w)}}, \tag{5a}$$

$$\mathcal{L}_{AO} = \sum_{(\hat{o}^i, \hat{a}^i, \hat{c}^i) \in \mathcal{B}} -\log \frac{\exp^{Sim(\hat{o}^i, \hat{a}^i)}}{\sum\limits_{\hat{o}^k \in \mathcal{B}} \exp^{Sim(\hat{o}^k, \hat{a}^i)}}, \tag{5b}$$

$$\mathcal{L}_{OAC}^{hn} = (\mathcal{L}_{OA}^{hn} + \mathcal{L}_{AO})/2, \tag{5c}$$

where $\mathcal{A}_{hn}^i$ denotes the attribute representation set of hard negatives for the object $O_i$, $Sim(\cdot, \cdot)$ measures the cosine similarity in a semantic space.

Similarly, Attribute-Category Contrastive (ACC) loss with the inclusion of hard negatives is formulated as follows:

$$\mathcal{L}_{AC}^{hn} = \sum_{(\hat{o}^i, \hat{a}^i, \hat{c}^i) \in \mathcal{B}} -\log \frac{\exp^{Sim(\hat{a}^i, \hat{c}^i)}}{\sum\limits_{\hat{c}^j \in \mathcal{B}} \exp^{Sim(\hat{a}^i, \hat{c}^j)} + \sum\limits_{\hat{c}^w \in \mathcal{C}_{hn}^i} \exp^{Sim(\hat{a}^i, \hat{c}^w)}}, \tag{6a}$$

$$\mathcal{L}_{CA}^{hn} = \sum_{(\hat{o}^i, \hat{a}^i, \hat{c}^i) \in \mathcal{B}} -\log \frac{\exp^{Sim(\hat{a}^i, \hat{c}^i)}}{\sum\limits_{\hat{a}^j \in \mathcal{B}} \exp^{Sim(\hat{a}^j, \hat{c}^i)} + \sum\limits_{\hat{a}^w \in \mathcal{A}_{hn}^i} \exp^{Sim(\hat{a}^w, \hat{c}^i)}}, \tag{6b}$$

$$\mathcal{L}_{ACC}^{hn} = (\mathcal{L}_{AC}^{hn} + \mathcal{L}_{CA}^{hn})/2, \tag{6c}$$

where $\mathcal{C}_{hn}^i$ denotes the category representation set of hard negatives for the object $O_i$.

As discussed in Section 2.2, it is hard to differentiate between category names in the representation space of LLMs. Inspired by the intra-modal contrastive loss to promote the model's ability to differentiate between hard nagative captions (Zhang et al., 2024a), we additionally define Category-Category Contrastive (CCC) loss as follows:

$$\mathcal{L}_{CCC} = \sum_{(\hat{o}^i, \hat{a}^i, \hat{c}^i) \in \mathcal{B}} -\log \frac{1}{\sum\limits_{\hat{c}^k \in \mathcal{C}_{hn}^i} \exp^{Sim(\hat{c}^i, \hat{c}^k)}}. \tag{7}$$

To maintain the generative power of the model, we use the attribute descriptions as LLM-augmented captiona to formulate the attribute description generation task. Therefore, the optimization object of the first stage can be defined as follows:

$$\mathcal{O}_{\beta,\theta}^{\text{I}} = \arg\min_{\beta,\theta} \mathcal{L}_G^{\text{att}} + (\mathcal{L}_{OAC}^{hn} + \mathcal{L}_{ACC}^{hn} + \mathcal{L}_{CCC})/2, \tag{8}$$

where $\mathcal{L}_G^{\text{att}}$ denotes the attribute description generation loss.

**Stage II: Classification-Centered Instruction Tuning.** In the second stage, we formulate the FGVR dataset as two kinds of instruction tuning data: open-set QA data and closed-set multiple-choice data. Then we fine-tune the model using this classification-centered instruction tuning data. Consequently, the optimization object of the second stage can be formulated as:

$$\mathcal{O}_{\beta,\theta}^{\text{II}} = \arg\min_{\beta,\theta} \mathcal{L}_G^{\text{cls}}, \tag{9}$$

where $\mathcal{L}_G^{\text{cls}}$ denotes the generation loss of classification-centered instruction tuning data.

Table 2: Comparison with leading methods on six FGVR datasets. #P denotes parameters count.

| Model | #P | Dog-120 | Bird-200 | Aircraft-102 | Flower-102 | Pet-37 | Car-196 | Avg. |
|---|---|---|---|---|---|---|---|---|
| LLaVA 1.5 | 7B | 38.96 | 35.24 | 34.71 | 51.37 | 52.25 | 46.92 | 43.24 |
| LLaVA-Next (Mistral) | 7B | 38.86 | 34.88 | 32.49 | 43.91 | 53.72 | 49.48 | 42.22 |
| MobileVLM v2 | 7B | 39.92 | 33.90 | 35.01 | 54.89 | 53.69 | 46.29 | 43.95 |
| InstructBLIP Vicuna | 7B | 41.60 | 32.78 | 31.68 | 50.90 | 54.92 | 48.25 | 43.36 |
| InstructBLIP Flan-T5-XL | 4B | 47.10 | 32.15 | 29.19 | 62.29 | 59.99 | 64.58 | 49.22 |
| Phi-3-Vision | 4B | 39.80 | 37.63 | 42.33 | 51.59 | 56.36 | 54.50 | 47.04 |
| BLIP2 Flan-T5-XL | 4B | 46.17 | 33.70 | 32.94 | 64.32 | 65.00 | 67.68 | 51.64 |
| InternLM XComposer 2 | 7B | 41.47 | 37.42 | 40.53 | 54.25 | 63.23 | 53.89 | 48.47 |
| Pali-Gemma | 3B | 51.68 | 36.62 | 39.87 | 69.64 | 75.42 | 64.64 | 56.31 |
| Idefics1 | 9B | 39.74 | 36.50 | 34.62 | 51.70 | 48.51 | 29.42 | 40.08 |
| Idefics2 | 8B | 57.96 | 47.17 | 56.23 | 72.78 | 81.28 | 80.25 | 65.95 |
| Qwen-VL-Chat | 10B | 66.18 | 52.30 | 45.96 | 75.95 | 87.82 | 76.23 | 67.41 |
| **Finedefics (ours)** | 8B | **72.86** | **57.61** | **63.82** | **89.88** | **92.18** | **84.67** | **76.84** |
| | | (+6.68) | (+5.31) | (+7.59) | (+13.93) | (+4.36) | (+4.42) | (+9.43) |

## 4 EXPERIMENTS

In this section, we evaluate the performance of Finedefics aiming to answer the following questions: (1) Can Finedefics effectively improve FGVR accuracy in MLLMs? (2) Does each core design of Finedefics benefit the accuracy improvement? (3) Is Finedefics effective in aligning visual objects and category names in the representation space of LLMs?

### 4.1 IMPLEMENTATION DETAILS

**Datasets.** We conduct experiments on several popular FGVR datasets that include CaltechUCSD Bird-200 (Wah et al., 2011), Stanford Car-196 (Krause et al., 2013), Stanford Dog-120 (Krause et al., 2013), Flower-102 (Nilsback & Zisserman, 2008), Oxford-IIIT Pet-37 (Parkhi et al., 2012), and FGVC-Aircraft (Maji et al., 2013). Following (Geigle et al., 2024), we leverage the test sets as resources for annotated data and frame FGVR as a multiple-choice task with well-defined answer candidates. To facilitate Finedefics, we select the training sets of them to construct attribute description, build open-set QA and closed-set multiple-choice instructing tuning data, ensuring that these images are different from the ones used in testing.

**Evaluated MLLMs.** We build Finedefics upon Idefics2 (Laurençon et al., 2024b) for its open-source accessibility and leading zero-shot performance. Several recent MLLMs of comparable parameter sizes are evaluated, including LLaVA 1.5 (Liu et al., 2024b), LLaVA-Next (Liu et al., 2024a), MobileVLM v2 (Chu et al., 2024), InstructBLIP Vicuna (Dai et al., 2024), InstructBLIP Flan-T5-XL (Dai et al., 2024), Phi-3-Vision (Abdin et al., 2024), BLIP2 Flan-T5-XL (Li et al., 2023), InternLM XComposer 2 (Dong et al., 2024), Pali-Gemma [1], Idefics1 (Laurençon et al., 2024a), Idefics2 (Laurençon et al., 2024b), and Qwen-VL-Chat (Bai et al., 2023).

**Training Settings.** All seeds are fixed across the training procedures for fairness. We train Finedefics using the QLoRa technique (Dettmers et al., 2024), updating adapters in the LLM and modality connector including perceiver resampler with 8 NVIDIA A6000 GPUs with 48G of memory. We use 4-bit quantization, with $\gamma = 8$ and $\alpha = 8$ for LoRA, and a learning rate of 2e-4. For both stages, the model is trained for one epoch with the warming steps of 60. The accumulated batch size is set to 64 and 128 for stage I and stage II, respectively.

### 4.2 MAIN RESULTS

In Table 2, we compare Finedefics with previous leading approaches on six popular FGVR datasets. Finedefics exhibits a significantly enhanced FGVR capability compared to a wide range of MLLMs. Notably, Finedefics shows superior performance than Idefics2 by an average of +10.89% and Qwen-VL-Chat of +9.43% across all datasets. Note that Finedefics is built upon Idefics2 (Laurençon et al., 2024b), a high-performing model in various vision-language and vision-centric tasks, and the

---

[1]https://ai.google.dev/gemma/docs/paligemma/model-card

Table 3: Analysis of Finedefics. "Original" represents the zero-shot performance of Idefics2.

(a) FT methods.

| Method | Avg. |
|---|---|
| Original | 65.95 |
| Finetune | 0.03 |
| **Finedefics (ours)** | **76.84** |

(b) Effectiveness of attributes.

| Method | Avg. |
|---|---|
| Original | 65.95 |
| CL (obj.-cat.) | 72.72 |
| **CL (obj.-att.-cat.)** | **76.84** |

(c) Training paradigm.

| Method | Avg. |
|---|---|
| Original | 65.95 |
| One stage | 25.42 |
| **Two stages** | **76.84** |

(a) Finetune      (b) CL (object-category)      (c) Finedefics (ours)

Figure 4: Representation visualization of Finetune, CL (object-category) and Finedefics.

enhanced performance on FGVR makes it a valuable foundation to benefit more advanced tasks with finer granularity.

### 4.3 ANALYSIS OF FINEDEFICS

**Does Fine-tuning Solely on Additional Open-set FGVR Data Bring Performance Gains?** In Table 3a, we fine-tuning Idefics2 solely on additional open-set FGVR data. We observe that it deteriorates the instruction following capability for answering multiple-choice questions in our test settings. Finedefics outperforms the fine-tuned model, which indicates that Finedefics effectively boosts FGVR accuracy by integrating attribute augmented alignment into the training paradigm rather than solely fine-tuning on additional data.

**Does Attribute Descriptions Contribute Performance Gains of Contrastive Learning?** To demonstrate the impact of augmenting contrastive learning with attribute descriptions, we conduct experiments and report the results in Table 3b. In the ablation experiments, we employ contrastive learning on object-category pairs without utilizing attribute descriptions. The results show that the attribute description benefits the alignment between visual objects and category names.

**Is Training in Two Stages Necessary for Building Finedefics?** We further analyze the necessity of training in two stages, i.e., representation alignment before instruction tuning. Specifically, we fine-tune Idefics2 with a combined loss of classification-centered instruction tuning and attribute augmented contrastive learning. The results are reported in Table 3c. We observe that fine-tuning in one stage is prone to struggle with optimization and leads to degraded performance, which indicates the effectiveness of training in two stages.

**Visualization - Does Finedefics Effectively Align Visual Objects and Category Names?** To substantiate our objective of enhancing the alignment between visual objects and category names with the auxiliary visual attributes, we randomly selected 100 data from Oxford-IIIT Pet-37 (Parkhi et al., 2012) for visualization. As illustrated in Figure 4a, a substantial gap between the object and category is observable in the data distribution when fine-tuning without contrastive learning. In Figure 4b, contrastive learning on object-category pairs without attribute descriptions involved fails to decrease the gap. In Figure 4c, with the usage of contrastive learning on object-attribute-category triples, the gap decreases significantly, thus boosting FGVR accuracy.

## 5 RELATED WORK

**Multi-modal Large Language Models**. Multimodal Large Language Models (MLLMs) aim to enhance machines' ability to understand and process complex information by integrating multiple data modalities such as vision, text, and audio. In recent years, MLLMs have achieved significant progress in three key areas. First was large-scale pre-training and fine-tuning, as seen in models like BLIP-2 (Li et al., 2023), LLaVA (Liu et al., 2024b), MiniGPT-4 (Zhu et al., 2023), PaLM-E (Driess et al., 2023), Kosmos-2 (Peng et al., 2023) and Visual ChatGPT (Wu et al., 2023), which used pre-training on vast multimodal datasets and were then fine-tuned for specific tasks, greatly improving the models' generalization ability and task performance. The second area was cross-modal consistency, focusing on ensuring information consistency across different modalities through techniques like contrastive learning. Models such as Shikra (Chen et al., 2023), FROMAGe (Koh et al., 2023), DLP (Jian et al., 2024), BuboGPT (Zhao et al., 2023b), ChatSpot (Zhao et al., 2023a), and Qwen-VL (Bai et al., 2023) enhanced performance in multimodal tasks by strengthening the alignment between modalities. The third area was interpretability and transparency. Models like ViperGPT (Surís et al., 2023), GPT-4 (Achiam et al., 2023), PandaGPT (Su et al., 2023), Video-LLaMA (Zhang et al., 2023a), and Video-ChatGPT (Maaz et al., 2023) enhanced the explainability of model decision-making by incorporating attention mechanisms and natural language feedback, enabling users to understand better and trust the model's output. Despite these achievements, MLLMs still face challenges, such as the inability to extract informative visual features, insufficient understanding of subordinate-level categories, and misalignment between visual objects and category names.

**Fine-Grained Visual Recognition**. FGVR (Welinder et al., 2010; Maji et al., 2013; Wei et al., 2021) aims to classify visually similar subordinate categories under a broader super-category, often requiring expert-provided auxiliary annotations (Krause et al., 2013; Zhang et al., 2014; Vedaldi et al., 2014; He & Peng, 2017) due to the subtle differences between objects. FGVR methods can be divided into three types: (i) attention-based methods enhance the model's ability to recognize subtle differences by focusing on the most critical areas of the image. (ii) hierarchical representation methods effectively handle the subtle differences between categories by constructing hierarchical feature representations that allow the model to refine image recognition progressively. (iii) metric learning methods improve the model's discriminative power in fine-grained classification tasks by learning a metric space where samples of the same class are closer and those of different classes are further apart. Moreover, TransHP (Wang et al., 2023) integrated vision-language models, making FGVR less reliant on annotations and adaptable across various tasks. HI2R (Chen et al.) introduced a hypergraph-guided approach that captures intra-class and inter-class relationships, enhancing the model's ability to discern subtle distinctions between fine-grained categories. CLEVER (Choudhury et al., 2024) extracted non-expert descriptions from images and trained a fine-grained textual similarity model to match image descriptions with Wikipedia document sentences accurately. Recent advancements like FineR (Liu et al., 2024c) employed large language models to translate visual attributes into text, enabling category identification without expert-defined labels.

## 6 CONCLUSION

In this paper, our objective is to analyze and boost the power of FGVR for MLLMs. We investigate the root cause of underperformance from three quintessential capabilities: object information extraction, category knowledge reserve, object-category alignment, we position the problem as the misalignment between visual objects and category names. To address the challenge, we propose Attribute Augmented Alignment, designed to use attribute descriptions as an intermediate point to bind them. Based on the aligned representation space, we build Finedefics, a new MLLM adept at identifying the subordinate-level category of the visual object. Our experiments, conducted on six popular FGVR datasets, demonstrate the remarkable performance of Finedefics. The validity of our methodology is substantiated through rigorous empirical studies.

**Future Works.** While Finedefics attains remarkable results across various FGVR datasets, it would encounter challenges in effectively learning new subordinate-level categories, and thus developing fine-tuning methods that can boost the continual FGVR capability for MLLMs is a promising future research direction.

## ACKNOWLEDGMENTS

This work was supported by the grants from the National Natural Science Foundation of China (61925201, 62132001, 62432001, 62373043) and Beijing Natural Science Foundation (L247006, 4252020).

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

## A APPENDIX

### A.1 MORE ABLATION STUDIES

**Attribute augmented alignment on other MLLMs.** We build Finedefics upon Idefics2 (Laurençon et al., 2024b). To confirm the general applicability of Finedefics, we conduct attribute augmented alignment ($A^3$) on another typical MLLM: LLaVA 1.5 (Liu et al., 2024b). As shown in Table 4, after employing our proposed method, LLaVA 1.5 gains an accuracy improvement by 13.97% on average, demonstrating the effectiveness and generalizability.

**Effects of attribute types.** We analyze the effects of specific attribute types in FGVR tasks. Specifically, we selectively remove typical attribute types from [color, shape, texture, size] to evaluate the contribution to performance improvement. As shown in Table 5, all four types of attributes play a crucial role in distinguishing subordinate-level categories, but the contribution varies with the dataset. For example, color and texture are more critical for specific datasets, like flowers and birds.

**Effects of hard negatives.** We compare using hard negatives and simple negatives for contrastive learning. Specifically, we replace hard negatives with randomly sampled simple negatives, meaning that the negatives used for contrastive learning are less visually similar to positives and easier to distinguish from them. As illustrated in Table 6a, after applying contrastive learning with simple negatives, the improvement is limited. With the utilization of hard negatives, the modality gap decreases further, and the model harvests a significant accuracy improvement.

**Effects of two-stage training stages.** We analyze the effects of two-stage training by evaluating Finedefics by selectively removing specific training processes within each stage. As shown in Table 6b, pretraining solely fails to follow the task instruction, while instruction tuning (I.T.) solely has a limited performance gain. Instead, pretraining and instruction tuning are complementary to further boost the accuracy, confirming the effectiveness of our two-stage training paradigm.

**Effects of description quality.** We first design an empirical study to evaluate the description quality, i.e, how reliable the attribute descriptions we built. Similar to the probing experiments in Section 2.2, we test the representation discriminability of our constructed attribute descriptions on the training set of Oxford-IIIT Pet-37 Parkhi et al. (2012) with a splitting ratio of 1:1. The accuracy is 68.27%, showing that the attribute descriptions can be well distinguished from each other though there exist subtle visual differences that are difficult to describe in words. Furthermore, to evaluate Finedefics's sensitivity to the description quality, we use three different quality levels of descriptions: (1) complete descriptions, (2) noisy descriptions (i.e., replacing some attribute descriptions with incorrect

Table 4: Effects of employing attribute augmented alignment ($A^3$) on LLaVA 1.5.

| Model | Dog-120 | Bird-200 | Aircraft-102 | Flower-102 | Pet-37 | Car-196 | Avg. |
|---|---|---|---|---|---|---|---|
| LLaVA 1.5 | 38.96 | 35.24 | 34.71 | 51.37 | 52.25 | 46.92 | 43.24 |
| LLaVA 1.5 + $A^3$ | **57.10** | **43.44** | **44.49** | **53.26** | **78.50** | **66.47** | **57.21** |

Table 5: Effects of attribute types.

| Color | Shape | Texture | Size | Dog-120 | Bird-200 | Aircraft-102 | Flower-102 | Pet-37 | Car-196 | Avg. |
|---|---|---|---|---|---|---|---|---|---|---|
| | ✓ | ✓ | ✓ | 71.15 | 57.40 | 61.63 | 88.19 | 91.36 | 84.22 | 75.66 |
| ✓ | | ✓ | ✓ | 72.49 | 57.75 | 62.02 | 90.19 | 90.73 | **85.26** | 76.41 |
| ✓ | ✓ | | ✓ | 71.98 | 55.47 | 60.82 | 88.81 | 89.48 | 80.70 | 74.54 |
| ✓ | ✓ | ✓ | | 70.65 | **57.84** | 59.74 | **91.49** | 90.30 | 82.94 | 75.49 |
| ✓ | ✓ | ✓ | ✓ | **72.86** | 57.61 | **63.82** | 89.88 | **92.18** | 84.67 | **76.84** |

Table 6: More ablation studies. "Original" represents the zero-shot performance of Idefics2.

(a) Negative types.

| Method | Avg. |
|---|---|
| Original | 65.95 |
| Simple Neg. | 74.26 |
| **Hard Neg. (ours)** | **76.84** |

(b) Training stages.

| Pretrain | I.T. | Avg. |
|---|---|---|
| ✓ | | 0.00 |
| | ✓ | 76.13 |
| ✓ | ✓ | **76.84** |

(c) Description quality.

| Method | Avg. |
|---|---|
| None | 72.72 |
| Noisy | 75.62 |
| **Complete (ours)** | **76.84** |

(d) Levels of detailed descriptions.

| Method | Avg. |
|---|---|
| None | 72.72 |
| Short | 76.11 |
| **Long (ours)** | **76.84** |

(e) Description construction.

| Method | Avg. |
|---|---|
| Upper-bound | 52.52 |
| Tag-based baseline | 50.48 |
| **Finedefics (ours)** | **51.12** |

ones by prompting ChatGPT OpenAI (2023)), and (3) no descriptions (i.e., object-category alignment w/o attribute). Results in Table 6c demonstrate Finedefics's robustness to description noise.

**Effects of levels of detailed descriptions.** To analyze the impact of description length and detail on alignment effectiveness, we compare three different levels of detailed descriptions: (1) long descriptions, (2) short descriptions (i.e., generating short descriptions of input images with BLIP-2 Li et al. (2023), and (3) no descriptions (i.e., object-category alignment w/o attribute). Results in Table 6d demonstrate that rich and informative category information expression plays a crucial role in boosting the alignment between visual objects and category names.

**Comparison with FineR.** We compare the clustering accuracy (cAcc) used in FineR Liu et al. (2024c) with the classification accuracy used in Finedefics. FineR can be considered as improving the FGVR performance by using the attributes as a zero-shot manner. Note that classification accuracy can be considered as the perfect case of Hungarian matching used to obtain clustering accuracy. Results in Table 7 show the superiority of building an attribute-aware model compared to a multi-agent system using attributes in a zero-shot manner.

**Effects of attribute description construction.** We evaluate the crucial rule of our proposed construction method acquiring per-sample attribute descriptions. To this end, we compare with an upper-bound and a baseline trained solely on Bird-200 dataset: (1) Upper-bound: we use Bird-200 (Wah et al., 2011)'s per-sample ground-truth attributes, annotated by humans, for pretraining and instruction tuning. (2) Tag-based baseline: Given the category name, we prompt ChatGPT (OpenAI, 2023) to acquire per-class attribute descriptions for each attribute tag used in our construction process, without using actual image samples. Then, these per-class sets of attribute-description pairs are assigned to each sample belonging to the class. Since for the same super category, the attribute values can differ significantly, this class-wise attribute description construction process introduces some noise. As shown in Table 6e, Finedefics surpasses the tag-based baseline, demonstrating the superiority of building detailed per-sample attribute descriptions. Moreover, we can observe that the

Table 7: Comparison with FineR. Clustering accuracy is used in FineR while classification accuracy is used in Finedefics.

| Model | Dog-120 | Bird-200 | Flower-102 | Pet-37 | Car-196 | Avg. |
|---|---|---|---|---|---|---|
| FineR | 48.10 | 51.10 | 63.80 | 72.90 | 49.20 | 57.00 |
| **Finedefics (ours)** | **72.86** | **57.61** | **89.88** | **92.18** | **84.67** | **79.44** |

Table 8: Object-category alignment quality for Idefics2 and Finedefics.

| Metric | Dog-120 | Bird-200 | Aircraft-102 | Flower-102 | Pet-37 | Car-196 | Avg. |
|---|---|---|---|---|---|---|---|
| Idefics2 | 0.14 | 0.12 | 0.12 | 0.09 | 0.15 | 0.18 | 0.13 |
| **Finedefics (ours)** | **0.28** | **0.17** | **0.30** | **0.28** | **0.28** | **0.32** | **0.27** |

accuracy of Finedefics is close to the upper-bound, showing the potential of using LLMs and VQA models to obtain large-scale attribute data for alignment without human annotations.

**Performance on common object recognition.** We investigate how Finedefics performs on common object recognition after pretraining and instruction tuning on FGVR datasets. As shown in Table 9, we test on ImageNet-adversarial (IN-adversarial) (Hendrycks et al., 2021b), ImageNet-rendition (IN-rendition) (Hendrycks et al., 2021a), and ImageNet-sketch (IN-sketch) (Wang et al., 2019). Results show that training solely on FGVR tasks has a minor impact on common object recognition accuracy. Therefore, training should be performed on a combined dataset comprising both coarse-grained and fine-grained recognition tasks. Developing fine-tuning methods that prevent such catastrophic forgetting to enhance FGVR capability without impacting common object recognition is a promising future research direction.

## A.2    MORE EVALUATION METRICS

**Confusion matrix analysis.** Besides the classification accuracy, we conduct confusion matrix analysis on Oxford-IIIT Pet-37 (Parkhi et al., 2012) to identify where the model struggles across categories. As shown in Figure 5, Finedefics can recognize most of the categories correctly, while struggles in identifying a small portion of categories, such as Wheaten Terrier (52%), Staffordshire Bull Terrier (58%), Ragdoll (71%), American Bulldog (74%), and Birman (78%).

**Alignment quality evaluation.** We provide a quantitative analysis of the object-category alignment quality for Idefics2 and Finedefics in Table 8. The object-category alignment quality is calculated as the mean cosine similarity between embeddings of visual objects and their corresponding category names of each class. We observe that Finedefics significantly increases the object-category alignment quality, which further demonstrates the effectiveness in boosting alignment.

## A.3    DISCRIMINABILITY COMPARISON BETWEEN DIFFERENT DATASETS

For each dataset, we provide a quantified analysis of inter-class distance and intra-class variance for visual objects, as well as inter-class distance for category names, respectively. As shown in Table 10, Dog-120 and Aircraft-102 have the smallest inter-class distance for visual objects, meaning that the subordinate-level categories of dogs and aircrafts are typically more visually similar and difficult to distinguish. Moreover, Pet-37 and Dog-120 have the largest intra-class distance for visual objects, meaning that attribute values differ significantly for the same subordinate-level category of pets. Most importantly, category names have lower discriminability than visual objects in the representation space despite the super category. For example, though Flower-102 has large inter-class distance and small intra-class variance for visual objects, the inter-class distance for category names is not significantly different from other datasets.

## A.4    QUALITATIVE COMPARISON

We visualize and analyze the predictions of Finedefics and Idefics2 in Figure 6. Finedefics successfully captures the nuance of the object features, setting them apart from visually similar subordinate categories. This confirms that Finedefics effectively captures fine-grained visual details

Table 9: Comparison on common object recognition datasets.

| Model | IN-adversarial | IN-rendition | IN-sketch | Avg. |
|---|---|---|---|---|
| Idefics2 | **79.84** | **93.23** | **68.21** | **80.43** |
| **Finedefics (ours)** | 75.96 | 92.43 | 66.77 | 78.39 |

Table 10: Discriminability comparison between different datasets. O and C denote visual object and category name, respectively.

| Metric | Dog-120 | Bird-200 | Aircraft-102 | Flower-102 | Pet-37 | Car-196 |
|---|---|---|---|---|---|---|
| Inter-class Dist. for O ($\uparrow$) | 0.36 | 0.37 | 0.31 | 0.41 | 0.43 | 0.49 |
| Intra-class Var. for O ($\downarrow$) | 0.27 | 0.20 | 0.12 | 0.07 | 0.28 | 0.19 |
| Inter-class Dist. for C ($\uparrow$) | 0.95 | 0.66 | 0.75 | 0.90 | 0.99 | 0.48 |

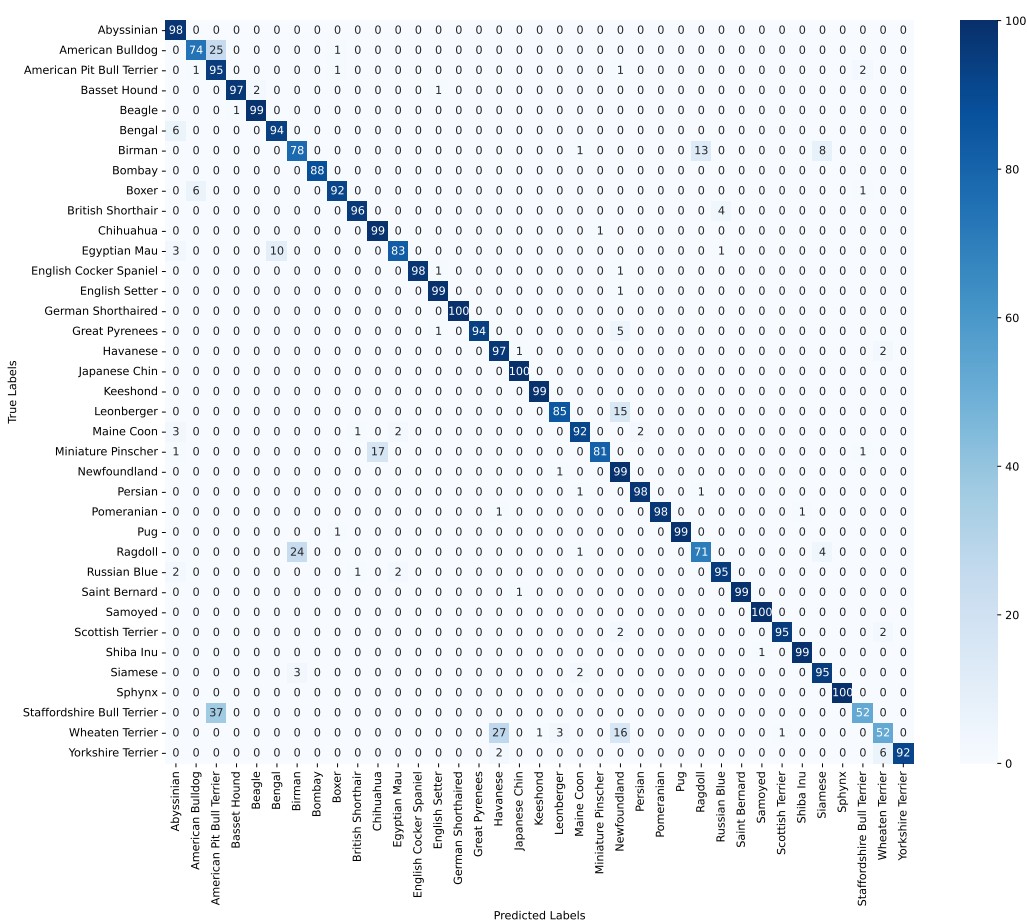

Figure 5: Confusion matrix of Oxford-IIIT Pet-37.

from images, connects them with category names in the representation space, and then generates precise, fine-grained predictions. Furthermore, Figure 7 shows two examples where Finedefics predicts incorrect labels. We randomly pick three examples from the incorrect label and observe that the ground-truth and predicted label share most of the attributes, with only a few exhibiting subtle differences. Concretely, as shown in the first row, Ragdolls are characterized by their fluffy, medium-to-long coats with distinct dark markings on the ears and around the eyes, whereas Siamese cats have short coats with darker color on their faces. Similarly, in the second row, pink primroses feature a light pink color with yellow-green centers, while tree mallows exhibit a vibrant pink color with dark purple to black centers.

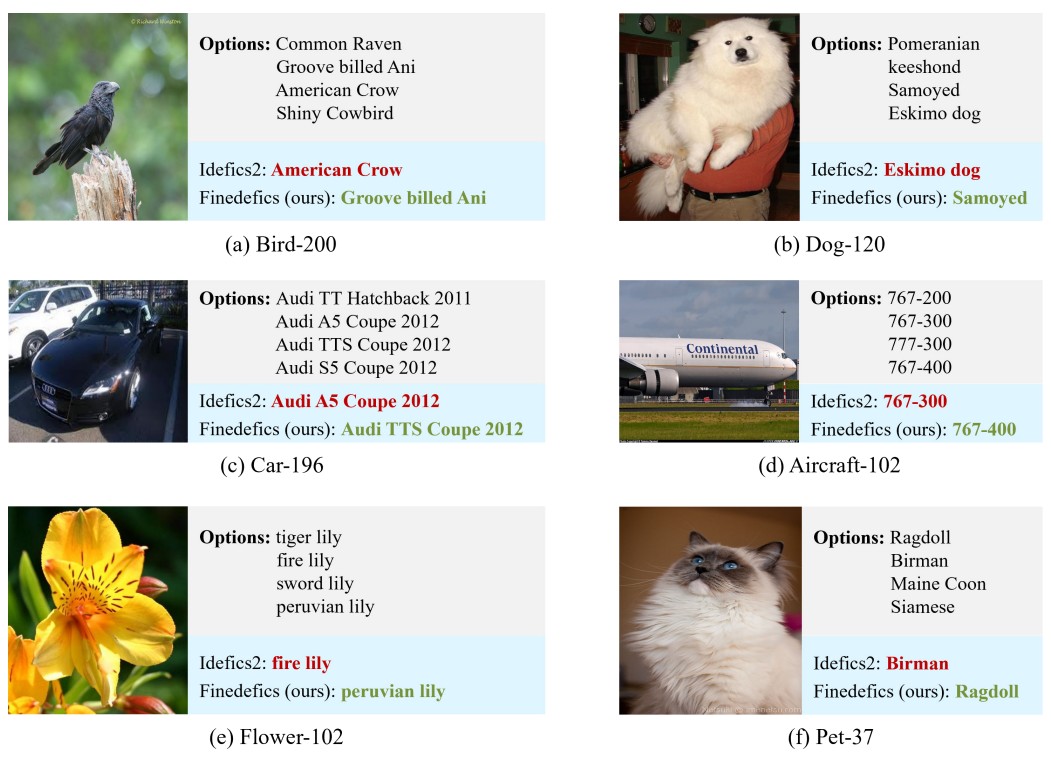

Figure 6: Qualitative comparison on FGVR datasets, where green indicates correct predictions and red indicates incorrect ones.

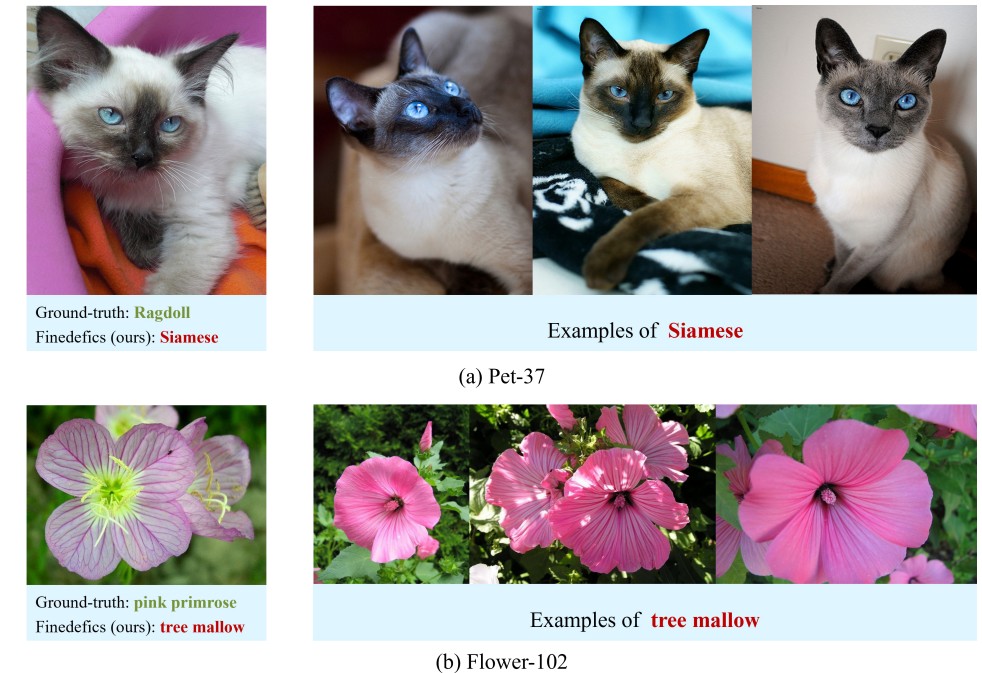

Figure 7: Error analysis examples. The left column shows the image for prediction, while the right column shows examples of the incorrect label.

