# OpenReview forum: "Analyzing and Boosting the Power of Fine-Grained Visual Recognition for Multi-modal Large Language Models"
_ICLR.cc/2025/Conference — ICLR 2025 Poster_

### Official Review · Reviewer_VD1i · 2024-10-17

**Soundness:** 4
**Presentation:** 3
**Contribution:** 4
**Rating:** 8
**Confidence:** 5

**Summary:**

This paper studies the task of Fine-grained Visual Recognition (FGVR). The authors point out the underperformance of modern MLLMs on FGVR and thoroughly explore the cause. The findings suggest that the misalignment between visual objects and category names is the main reason that hinders MLLMs' performance on FGVR. To this end, a novel instruction-tuning framework named Finedeficsfor is proposed to enhance the performance of MLLMs on FGVR by leveraging attribute descriptions. Extensive comparisons, experiments, and ablation studies are conducted across six FGVR benchmarks to validate the effectiveness of the proposed method (model) and its components.

**Strengths:**

**Before listing the strengths of this paper point by point, I would like to first acknowledge and highlight the research approach employed by this work.**
```
The way this paper conducts its research on improving MLLMs’ capabilities for FGVR demonstrates the authors' rigorous and meticulous scientific approach: identifying the problem (MLLM underperformance in FGVR) → understanding the cause (misalignment between objects and categories) → proposing a solution (leveraging informative attribute descriptions to effectively align visual objects and category names in the representation space of LLMs) → achieving improved performance (surpassing Idefics2 and Qwen-VL-Chat by an average of +10.89% and +9.43%, respectively).

I greatly appreciate this work's research methodology. The carefully designed analysis of MLLMs in FGVR presented in Section 2 will provide valuable insights and guidance for the community and future researchers. These insights may be even more valuable than the proposed method itself and its performance improvements, as they will significantly aid further research in this area. Thank you to the authors for the contribution!
```

**Strengths Point-by-Point:**
- The preliminary exploration of revisiting MLLMs’ capabilities for FGVR shown in Figure 1 is really interesting and intuitive. **I really appreciate this experimental design**. It highlights the core issue—Object-Category Alignment—that hinders MLLMs' FGVR performance.
- The proposed instruction tuning method, consisting of Attribute-Category Contrastive, Object-Attribute Contrastive, and Category-Category Contrastive, is novel and well-designed. The design is both technically innovative and conceptually intuitive.
- The proposed method boosts MLLM performance by a significant margin and consistently outperforms other MLLMs at similar parameter scales.
- Comprehensive ablation studies are conducted to assess the effectiveness of the proposed components.

**Weaknesses:**

(1) Since the proposed method relies on the attributes obtained via FineR [1] cross-modal chain-of-thoughts prompting,  FineR's FGVR performance should be compared as a baseline in the main experiments. FineR can be consider as improving VLM's FGVR performance by using the attributes as a zero-shot manner. The clustering accuracy used in FineR can be compared with classification accuracy (classification accuracy can be consiered as the perfect case of Hungarian matching used to obtain clustering accuracy). Although an apple-to-apple comparison is hard since FineR consists of multiple agents, this comparison can serve as a good reference to the readers.

(2) In the study of Finedefics in Section 4.3, the influence of construction of Attribute Descriptions should also be examined since it is a crucial component for acquiring per-sample, per-class attribute descriptions. To this end, one experimental design I can suggest is comparing the current construction method with an upper-bound and a baseline on a dataset, such as CUB-200:
  i) **Upper-bound:** The authors can use CUB-200’s ground-truth attributes, annotated by humans, for instruction tuning.
  ii) **Tag-based baseline:** Given the class names (e.g., Blue-throated Blue Warbler), the authors can directly prompt an LLM to acquire both useful attributes (e.g., body color pattern) as tags and possible attribute descriptions (e.g., dark blue) for each class’s attribute tags, without using actual image samples. Then, these per-class sets of attribute-description pairs can be assigned to each sample belonging to the class. Since this acquisition process is not conditioned on the training images, it might be noisier. It would be interesting to see how sensitive the model performance is to this upper-bound and baseline.

(3) The proposed method requires instruction tuning of MLLMs. Although the main goal is to improve MLLM performance for FGVR, it is still important to investigate whether this FGVR-centric tuning hampers the performance of MLLMs on general QA or common object recognition.

**Minor:**
- At P4#L193: Similarity —> Similarly

**P.S.:** I understand that the rebuttal period provides limited time to address all the concerns and questions raised. For me, the most critical issues are Q1 (no training required) and Q2 outlined above. **If time constraints prevent the authors from addressing all my concerns, please prioritize responding to Q(1) and Q(2), and allocate time to address the other reviewers' major concerns.** It is fine for me if the authors could not address all my concerns and questions (which might take time to train) within the limited time during the rebuttal period. No worries. But it is recommended to include them in the final revision.

[1] Liu, M., Roy, S., Li, W., Zhong, Z., Sebe, N., & Ricci, E. (2024). Democratizing fine-grained visual recognition with large language models.  In ICLR, 2024.

**Questions:**

(1) In the current paper, pet (dogs and cats) images are largely used as experimental subjects in the analysis of MLLMs in FGVR in Section 2. Pets, such as dogs and cats, have a unique characteristic: for the same pet category (or breed), the attribute values (e.g., “Fur Color: Brown”, “Fur Pattern: Dots”) can differ significantly (the so-called large intra-class variance in FGVR). This characteristic also applies to cars (StanfordCar196). However, I wonder: what would be the observation for birds (CUB200) or flowers (Oxford-flower)? For the same bird or flower species, the attribute values are rather fixed—i.e., the same bird species will always have the same body color pattern. In this case, would the observations and conclusions change? Would the category names still have lower discriminability in the representation space?

(2) Can the authors provide: 1) qualitative results; 2) failure case analysis? It is necessary to show how the model behaves towards FGVR QA since this is the way it will be used in deployment.

(3) One open question: Why is instruction-tuning MLLMs for FGVR useful and valuable? If the goal is to achieve high FGVR performance, supervising a strong pre-trained model on FGVR datasets can achieve higher performance. Since both paradigms require training, why do we use MLLMs for FGVR?

**Further Suggestions:**
- For the mining of hard negatives for each category, a suggestion is to explore using negatives that are highly similar to the target class but dissimilar in diverse attribute aspects. The current method relies on overall CLIP similarity for decision-making. Note that this is just a suggestion—I understand that it takes time to implement and train the model. The authors are not required to show this during the rebuttal period, but it would be interesting to see.

---

> ### Author Response · Authors · 2024-11-25
> **Answers to the questions raised in review**
>
> We thank reviewer VD1i for their remarks and their thoughtful view. We answer the questions below:
>
> **Q1: Comparison with FineR**
>
> We compare the clustering accuracy (cAcc) used in FineR with the classification accuracy used in Finedefics. With the former denoting FineR and the latter denoting Finedefics, the results are: a) Dogs-120 (48.10 vs 72.86), b) Bird-200 (51.10 vs 57.61), c) Flowers102 (63.80 vs 89.88), d) O.-Pet (72.90 vs 92.18), e) S.-Cars (49.20 vs 84.67). This shows the superiority of building an attribute-aware model compared to a multi-agent system using attributes in a zero-shot manner.
>
> **Q2: Effects of attribute description construction**
>
> We evaluate the crucial rule of our proposed construction method acquiring per-sample attribute descriptions. To this end, we compare with an upper-bound and a tag-based baseline trained solely on Bird-200 dataset. Results (Upper-bound: 52.52, Tag-based baseline: 50.48, Ours: 51.12) show that Finedefics surpasses the tag-based baseline, demonstrating the superiority of building detailed per-sample attribute descriptions. Moreover, we can observe that the accuracy of Finedefics is close to the upper-bound, showing the potential of using LLMs and VQA models to obtain large-scale attribute data for alignment without human annotations.
>
> **Q3: Performance on common object recognition**
>
> We investigate how Finedefics performs on common object recognition after pretraining and instruction tuning on FGVR datasets. The results show that training solely on FGVR tasks has a minor impact on common object recognition accuracy: a) ImageNet-adversarial (79.84 vs 75.96), b) ImageNet-rendition (93.23 vs 92.43), and c) ImageNet-sketch (68.21 vs 66.77). Therefore, training should be performed on a combined dataset comprising both coarse-grained and fine-grained recognition tasks. Developing fine-tuning methods that prevent such catastrophic forgetting to enhance FGVR capability without impacting common object recognition is a promising future research direction.
>
> **Q4: Discriminability comparison between different datasets**
>
> For each dataset, we provide a quantified analysis of intra-class variance for visual objects, as well as inter-class distance for category names, respectively. O.-Pet and Dogs-120 have the largest intra-class distance for visual objects (0.28 and 0.27), meaning that attribute values differ significantly for the same subordinate-level category of pets. Most importantly, category names have lower discriminability than visual objects in the representation space despite the super category. For example, though Flowers102 has large inter-class distance (0.41) and small intra-class variance for visual objects  (0.07), the inter-class distance for category names (0.90) is not significantly different from other datasets.
>
> **Q5: Qualitative comparison and error analysis**
>
> We provide qualitative comparison and error analysis examples in Figure 6 and 7 in Appendix, respectively. Finedefics successfully captures the nuance of the object features, setting them apart from visually similar subordinate categories. This confirms that Finedefics effectively captures fine-grained visual details from images, connects them with category names in the representation space, and then generates precise, fine-grained predictions.
>
> **Q6: MLLMs for FGVR**
>
> Though supervising a strong pre-trained discriminative model (i.e., VLMs) on FGVR datasets can achieve high performance, there exist three advantages of using generative models (i.e., MLLMs):
>
> (1) VLMs can only be used in a closed-world setting where the label set is known, while MLLMs can be used in an open-world setting where the label set is not provided and a closed-world setting where classes are concatenated in the prompt [a].
>
> (2) Due to the characteristic of discriminative models, VLMs encounter severe catastrophic forgetting in class-incremental learning scenarios. Nevertheless, MLLMs can harness the wealth of semantic correspondences between texts and images to alleviate catastrophic forgetting, while there’s no requirement to expand the classifier with each new category, unlike in the case of VLMs [b].
>
> (3) VLMs can only be supervised to perform FGVR task. However, recognizing an object is a prerequisite for answering complex questions about it.  We attempt to further transfer the enhanced FGVR performance of MLLMs into its general capabilities, such as object-centric, knowledge-intensive question answering, where the question can only be accurately answered if the class of the object is correctly identified [a].
>
> [a] Zhang et. al., Why are Visually-Grounded Language Models Bad at Image Classification?  Arxiv 2024.
>
> [b] Cao et. al., Generative Multi-modal Models are Good Class-Incremental Learners, CVPR 2024.

---

> ### Comment · Reviewer_VD1i · 2024-11-25
> **Response to the Authors - Good Paper and Study**
>
> Dear Authors,
>
> Thank you for the detailed, precise rebuttal. I have checked the author rebuttal and the revised paper. All my questions are well-addressed. I believe, with new, deeper investigation requested by me and other reviewers, the current revised paper has been significantly strengthend. I have also checked the Questions & Answers from my peer reviewers. The Authors did a really good job in the rebuttal. Kuods to the authors. The current revised paper looks really good.
>
> Therefore, I decide to insist my initial `8: accept` rating, while keep `5 confidence`. This is a good paper with novel method, comprehensive experiments and ablations, where both the method and experiments present important contribution to FGVR field.
>
> Good luck!
>
> Best regards,
>
> Reviewer VD1i

---

> > ### Author Response · Authors · 2024-11-26
> > **Thanks for the reply**
> >
> > Thank you so much for your reply. We sincerely appreciate your constructive comments and suggestions, which greatly enhance the quality of our paper. We will incorporate all these additional results into the final version of the paper.

---

### Official Review · Reviewer_1pZP · 2024-10-30

**Soundness:** 3
**Presentation:** 2
**Contribution:** 2
**Rating:** 3
**Confidence:** 4

**Summary:**

The paper proposes a model called Finedefics to improve FGVR in MLLMs, which typically struggle with distinguishing subordinate-level visual categories. The authors identify misalignment between visual objects and category names as a key challenge, proposing a two-stage training approach that uses attribute-augmented contrastive learning to align objects and category descriptions more effectively. Evaluations across several FGVR datasets show that Finedefics significantly improves recognition accuracy over existing MLLMs.

**Strengths:**

1. The paper does an excellent job identifying the specific challenges of FGVR within MLLMs. It is worth mentioning that this paper provides valuable insights into the object-category misalignment issues in existing MLLMs.
2. The proposed Finedefics framework introduces a creative method for addressing FGVR challenges by employing attribute-augmented alignment.
3. The paper’s methodology is thorough, with a well-delineated two-stage training process.

**Weaknesses:**

In general, the motivation for this submission is easy to understand and insight is interesting, but the experimental analysis is limited. In addition, there are still several weaknesses, as follows:
1. The proposed Finedefics method was constructed and validated solely based on the Idefics2 model, with no tests conducted on other prominent MLLMs. Different MLLMs, like BLIP-2, LLaVA, or Qwen-VL-Chat, vary significantly in their abilities to process visual features, generate textual descriptions, and perform contrastive learning. Without experimental validation across these models, it’s challenging to confirm the general applicability of Finedefics.
2. The paper contrasts "object-category" and "object-attribute-category" pairs, but it does not analyze the effects of specific attribute types (e.g., color, shape, texture). An ablation study could selectively remove particular attribute types to evaluate their contribution to performance improvement. This would help clarify which attributes are most important in fine-grained recognition tasks, particularly if certain attributes are more critical for specific datasets, like flowers or birds.
3. While standard contrastive learning ablation is provided, the study lacks a comparison between hard negatives and simple negatives. Additional experiments could assess performance when hard negatives are removed, highlighting whether they significantly enhance alignment and fine-grained recognition capability.
4. The current study compares one-stage and two-stage training, but more granular analysis could be performed. For example, selectively adding or removing specific training processes within each stage could reveal the individual effects of the alignment and instruction-tuning stages. This would help clarify the extent to which each stage contributes and whether they are complementary under certain conditions.
5. The methodology relies on attribute extraction from pre-trained models like GPT-4 and LLaVA. However, there’s no empirical analysis of how reliable or accurate these attributes are for all instances in the datasets, especially for categories with subtle visual differences. Since the attribute descriptions are generated by pre-trained LLM and VQA models, it would be valuable to conduct ablation studies using different quality levels of descriptions (e.g., complete descriptions, noisy descriptions, or no descriptions) to test the model’s robustness and dependency.
6. The paper mainly uses accuracy as an evaluation metric, which may not capture the nuanced performance differences in FGVR tasks. Metrics like confusion matrix analysis could provide more insight, especially for identifying where the model struggles across categories. Furthermore, the visualization section could include quantitative measures to assess the object-category alignment quality, rather than relying solely on visual interpretation.
7. The paper suggests that category names alone lack discriminability within MLLMs. A test could be designed where category names are replaced with varying levels of detailed descriptions (e.g., short versus long descriptions) to analyze the impact of description length and detail on alignment effectiveness. This would verify whether the form of category information expression significantly affects model performance.

**Questions:**

In addition to the issues mentioned in Weaknesses, there is another point  about generalization  that needs to be clarified, as follows:
Each MLLM has different levels of perceptual sensitivity to fine-grained details. For example, some models excel in low-level visual feature recognition, while others are better suited for high-level semantic understanding. The proposed approach relies heavily on the specific architecture of Idefics2, particularly the model’s vision encoder, modality connector, and alignment layers. Other MLLMs might not have identical modules or configurations, which could make direct application of the proposed training paradigm challenging.

---

> ### Author Response · Authors · 2024-11-25
> **Answers to the questions raised in review**
>
> We thank reviewer 1pZP for their remarks and their thoughtful view. We answer the questions below:
>
> **Q1: Attribute augmented alignment on other MLLMs**
>
> To confirm the general applicability of Finedefics, we conduct attribute augmented alignment on another typical MLLM: LLaVA 1.5. After employing our proposed method, LLaVA 1.5 gains an accuracy improvement by 13.97% on average (43.24 vs 57.21), demonstrating the effectiveness and generalizability. We will include results based on more MLLMs in the final revision.
>
> **Q2: Effects of attribute types**
>
> We analyze the effects of specific attribute types in FGVR tasks. Specifically, we selectively remove typical attribute types from [color, shape, texture, size] to evaluate the contribution to performance improvement. All four types of attributes play a crucial role in distinguishing subordinate-level categories, but the contribution varies with the dataset (see Table 5 in Appendix). For example, color and texture are more critical for specific datasets, like flowers and birds.
>
> **Q3: Effects of hard negatives**
>
> We compare using hard negatives and simple negatives for contrastive learning. Specifically, we replace hard negatives with randomly sampled simple negatives, meaning that the negatives used for contrastive learning are less visually similar to positives and easier to distinguish from them. After applying contrastive learning with simple negatives, the improvement is limited (65.95 vs 74.26). With the utilization of hard negatives, the modality gap decreases further, and the model harvests a significant accuracy improvement (74.26 vs 76.84).
>
> **Q4: Effects of two-stage training stages**
>
> We analyze the effects of two-stage training by evaluating Finedefics by selectively removing specific training processes within each stage. Pretraining solely fails to follow the task instruction (0.0), while instruction tuning solely has a limited performance gain (65.95 vs 76.13). Instead, pretraining and instruction tuning are complementary to further boost the accuracy (76.13 vs 76.84), confirming the effectiveness of our two-stage training paradigm.
>
> **Q5: Effects of description quality**
>
> We first design an empirical study to evaluate the description quality, i.e, how reliable the attribute descriptions we built. Similar to the probing experiments in Section 2.2, we test the representation discriminability of our constructed attribute descriptions on the training set of Oxford-IIIT Pet-37 with a splitting ratio of 1:1. The accuracy is 68.27%, showing that the attribute descriptions can be well distinguished from each other though there exist subtle visual differences that are difficult to describe in words. Furthermore, to evaluate Finedefics’s sensitivity to the description quality, we use three different quality levels of descriptions: (1) complete descriptions, (2) noisy descriptions (i.e., replacing some attribute descriptions with incorrect ones by prompting ChatGPT), and (3) no descriptions (i.e., object-category alignment w/o attribute). Results (complete: 76.84, noisy: 75.62, none: 72.72) demonstrate Finedefics’s robustness to the description noise.
>
> **Q6: More evaluation metrics**
>
> (1) Confusion matrix analysis: Besides the classification accuracy, we conduct confusion matrix analysis on Oxford-IIIT Pet-37 to identify where the model struggles across categories (see Figure 5 in Appendix). Finedefics can recognize most of the categories correctly, while struggles in identifying a small portion of categories, such as Wheaten Terrior (52%), Staffordshire Bull Terrier (58%), Ragdoll (71%), American Bulldog (74%), and Birman (78%).
>
> (2) Alignment quality evaluation: We provide a quantitative analysis of the object-category alignment quality for Idefics2 and Finedefics. The object-category alignment quality is calculated as the mean cosine similarity between embeddings of visual objects and their corresponding category names of each class. We observe that Finedefics significantly increases the object-category alignment quality (0.13 vs 0.27), which further demonstrates the effectiveness in boosting alignment.
>
> **Q7: Effects of levels of detailed descriptions**
>
> To analyze the impact of description length and detail on alignment effectiveness, we compare three different levels of detailed descriptions: (1) long descriptions, (2) short descriptions (i.e., generating short descriptions of input images with BLIP-2, and (3) no descriptions (i.e., object-category alignment w/o attribute). Results (long: 76.84, short: 76.11, none: 72.72) demonstrate that rich and informative category information expression plays a crucial role in boosting the alignment between visual objects and category names.
>
> **Q8: Generalization**
>
> In fact, our method does not rely on the specific architecture of Idefics2, since it conducts attribute augmented alignment on the output embeddings of LLM and the architectures of existing MLLMs are built upon LLMs.

---

> > ### Comment · Reviewer_1pZP · 2024-11-27
> > **Response to author’s rebuttal**
> >
> > Thank you for your rebuttal. Unfortunately, it does not fully address my concerns. I find the proposed method to have limited novelty and scalability, and I remain skeptical about its claimed effectiveness and whether it is truly open-sourced. Moreover, given the strong performance of existing large-scale vision-language models in fine-grained recognition, the added value of this approach is diminishing. Therefore, I will maintain my original score.

---

> > > ### Author Response · Authors · 2024-11-28
> > > **Further discussion**
> > >
> > > Thank you for your response. I would like to provide further clarification for each of your points below.
> > >
> > > **(1) Insight and novelty**: To the best of our knowledge, our work is **the first attempt to position the bottleneck of MLLMs' fine-grained visual recognition (FGVR) capability**: the misalignment between visual objects and category names, **providing valuable insights and guidance for the MLLM community and future researchers**. This indicates that **purely optimizing representations of visual encoders or scaling LLMs will lead to limited improvement compared to boosting the alignment quality.** For example, changing the ViT-L/14-224 image encoder to a higher resolution (336) only yields **~1%** better performance, scaling MobileVLM v2 from 3B to 7B with otherwise identical training leads to only a marginal **~4%** accuracy gain [a], while we see gains of **10.89%** by improving object-category alignment quality of Idefics2.
> > >
> > > **(2) Value of our performance gain**: Prior efforts [a,b] mentioned that **existing MLLMs struggle for FGVR, which is far from satisfactory**. Since FGVR serves as foundations for more advanced capabilities such as object-centric, knowledge-intensive question answering, where the question can only be accurately answered if the class of the object is correctly identified [a], **the value of our performance gain is far beyond FGVR**. For example, suppose a virtual assistant is helping agronomists diagnosing pests. In that case, the model must correctly identify the pest species to answer questions like "How to treat the pests?"
> > >
> > > **(3) Open-source**: Due to the double-blind policy, **we will release our code, model, and data immediately once the policy allows**.
> > >
> > > Please let us know if any additional questions require further clarification during the extended discussion period.
> > >
> > > [a] Geigle et. al., African or European Swallow? Benchmarking Large Vision-Language Models for Fine-Grained Object Classification, Arxiv 2024.
> > >
> > > [b] Zhang [Stanford University] et. al., Why are Visually-Grounded Language Models Bad at Image Classification? NeurIPS 2024.

---

### Official Review · Reviewer_qkG7 · 2024-11-02

**Soundness:** 3
**Presentation:** 3
**Contribution:** 2
**Rating:** 5
**Confidence:** 3

**Summary:**

This research paper investigates the challenges faced by multi-modal large language models (MLLMs) in fine-grained visual recognition (FGVR), a task involving identifying specific sub-categories within broader categories, such as distinguishing different types of birds. The authors pinpoint the root cause of this underperformance as a misalignment between visual object representations and corresponding category names within the model's representation space. To address this, they propose Finedefics, an MLLM architecture that uses informative attribute descriptions of objects as an intermediate point to bridge this gap. This approach effectively aligns visual objects and category names during training, leading to significant performance gains in FGVR tasks compared to existing models.

**Strengths:**

- Section 2 provides a thorough and engaging explanation of the problem.
- Problem formulation is straightforward and easy to understand.
- The paper demonstrates significant improvements across all tested datasets.

**Weaknesses:**

- Despite the strong results, the proposed methods lack substantial novelty compared to prior works ([1], [2]) that also leverage foundation models for data augmentation and model fine-tuning.

**Questions:**

- In the experiment in Section 2.1, what prompts did you use? Also, to clarify, was linear probing performed after the connector or the LLM?
- Could you explain Figure 2 (c+f) in more detail?

- [1] H. Laurençon, L. Tronchon, M. Cord, and V. Sanh. *What matters when building vision-language models?*
- [2] M. Yuksekgonul, F. Bianchi, P. Kalluri, D. Jurafsky, and J. Zou. *When and Why Vision-Language Models Behave like Bags-of-Words, and What to Do About It?*

---

> ### Author Response · Authors · 2024-11-25
> **Answers to the questions raised in review**
>
> We thank reviewer qkG7 for their remarks and their thoughtful view. We answer the questions below:
>
> **Q1: Novelty compared to prior works**
>
> Comparing [1] and our method, we both use hard negatives to enhance the performance of contrastive learning.  Nevertheless, contrastive learning solely on image-caption pairs with hard negatives [1] fails to fully align visual objects and category names (65.95 vs 72.72), as shown in Table 3(b) in the manuscript. In contrast, we propose an attribute augmented contrastive learning paradigm to better alleviate the misalignment between visual objects and category names, further boosting the accuracy by a large margin (72.72 vs 76.84).
>
> Comparing [2] and our method, we build upon the proposed Idefics2 [2] but is specifically tailored to enhance its FGVR capability. Despite the dedicated design of multi-stage pretraining and instruction fine-tuning on large-scale data [2], alignment in the distribution of visual objects and category names is still unattainable, leading to underperformance in FGVR tasks (65.95 on average). In contrast, we propose a FGVR-aware training paradigm comprising attribute augmented pretraining and classification-centered instruction tuning, improving FGVR accuracy effectively (76.84 on average).
>
> **Q2: Experimental details in Section 2.1**
>
> We feed the image token sequence into the LLM without appending any prompts. For MLLM (i.e., Idefics2), linear probing is performed after the LLM.  For visual encoder (i.e., SigLiP), linear probing is performed before the connector.
>
> **Q3: Detailed explanation about Figure 2 (c+f)**
>
> We randomly sample 120 object-category pairs from Stanford Dog-120. For Idefics2, we obtain the last token embedding of the object image $\hat{o} _ {m}^{i}$ and the category name $\hat{c} _ {n}^{i}$ after the LLM. For SigLIP, we obtain projected ${[CLS]}$ embedding from the last layer of visual encoder $\hat{o} _ {\text{CLS}}^{i}$ and text encoder $\hat{c} _ {\text{CLS}}^{i}$. We normalize $\hat{o} _ {m}^{i}$ , $\hat{c} _ {m}^{i}$ for t-SNE visualization in the same representation space, shown in Figure 2(f). Similarly, we normalize $\hat{o} _ {\text{CLS}}^{i}$, $\hat{c} _ {\text{CLS}}^{i}$ for t-SNE visualization in the same representation space, shown in Figure 2(c).

---

### Author Response · Authors · 2024-11-25
**New revision**

We want to thank the reviewers for their thoughtful comments and questions, and we feel the subsequent additions make the paper much stronger and sound. We have updated the manuscript with a new revision, adding appendices with more quantitative and qualitative evidence to support our answers in the discussions with reviewers.

---

### Comment · Reviewer_VD1i · 2024-11-25
**A Kind Reminder to Peer Reviewers from Reviewer VD1i: The authors have provided a well-done rebuttal and a revised paper, including very interesting experiments.**

Dear Peer Reviewers,

I am writing this message to kindly encourage you to check the rebuttal and revised paper provided by the authors.

I have reviewed the rebuttal responses and the revised paper submitted by the Authors. The Authors have conducted extensive and insightful experiments, accompanied by thorough discussions, to investigate the influence of the following aspects on their proposed method:
1. Attribute augmented alignment on other MLLMs.
2. Effects of attribute types.
3. Effects of hard negatives.
4. Effects of two-stage training stages.
5. Effects of description quality.
6. Effects of levels of detailed descriptions.
7. Comparison with FineR. (suggested in my review)
8. Effects of attribute description construction. (suggested in my review)
9. Performance on common object recognition. (suggested in my review)

All my suggestions and questions have been thoroughly addressed. I am impressed by the comprehensiveness, effort, and quality of both the rebuttal and the revised paper. The current revision appears satisfactory to me. It is indeed not easy to complete all these studies during the limited time given by the rebuttal period.

Therefore, I woud like to kindly encourage the peer reviewers to check if your concerns and questions have been adequately addressed in the author rebuttal and, if necessary, engage in further discussion with the authors.

Thank you!

Best regards,

Reviewer VD1i

---

### Meta-Review · Area_Chair_4thb · 2024-12-19

**Metareview:**

This paper studies the task of Fine-grained Visual Recognition (FGVR) with Multi-modal Large Language Models (MLLM). The authors conducted several experiments to disclosure the main reason that restricted the performance of MLLM for FGVR. Based on this finding, the authors introduce a two stage fine-tuning framework by leveraging attribute descriptions. Extensive experiments show the effectiveness of the proposed method in MLLM for FGVR.

The paper was reviewed by three reviewers.

The strengths of the paper include: 1) Identifying the specific challenges of FGVR within MLLMs (acknowledged by all reviewers); 2) introduce a creative/technically innovative method for solve the drawback of MLLMs (acknowledged by  Reviewer 1pZP and Reviewer VD1i); and 3) strong performance on all datasets (acknowledged by all reviewers).

Initially, the reviewers raised many comments and concerns, mainly include: 1) novelty compared with [1, 2] (Reviewer qkG7) ; 2) detailed descriptions of the method (Reviewer qkG7); 3) more experiments compared to different variants (Reviewer 1pZP and Reviewer VD1i); and 4) value of tuning MLLMs for FGVR compared traditional supervised training on VLM (Reviewer VD1i).

The authors have provided a rebuttal. Both Reviewer 1pZP and Reviewer VD1i have attended the discussion-phase.

- Reviewer VD1i is happy to the response of the authors and finally kept the high positive score (8). In addition, Reviewer VD1i would like to champion the acceptance of this paper.

- Reviewer qkG7 did not attend the discussion-phase. The AC has carefully checked the response of the authors and found that the concerns of Reviewer qkG7 have been well solved, especially the concern of the novelty. The AC agrees that the proposed method is specifically tailored to enhance FGVR capability of MLLM, in which attribute augmented contrastive learning paradigm is effective and has not been considered in [1, 2].

- Reviewer 1pZP has attended the discussion-phase but raised several further concerns. 1) novelty and scalability; 2) open-source; 3) added value (or effectiveness) compared to existing VLM models. The authors have provided further responses to these concerns while Reviewer 1pZP did not further reply to the authors. The AC has checked the responses of the authors and agrees that the further concerns have been solved by the authors. Note that, Reviewer 1pZP has already confirmed the novelty of the method in the initial comments (i.e., creative method). This is somewhat disagree with his/her further concern. The authors also promised to release the code. Finally, existing MMLM did not show very strong performance in FGVR while the proposed method achieves a strong improvements, showing the effectiveness of the proposed method.

Given the above considerations, the AC thinks 1) the authors have well-solved the concerns of all reviewers and 2) this work provides a new insight and new effective framework for the task of FGVR within MLLM. To this end, the AC would like to recommend acceptance to this paper. If accepted, the AC strongly encourages the authors to include all the discussions and experiments in the final version and release the source code along with the trained-models as promised.s


[1] H. Laurençon, L. Tronchon, M. Cord, and V. Sanh. What matters when building vision-language models?

[2] M. Yuksekgonul, F. Bianchi, P. Kalluri, D. Jurafsky, and J. Zou. When and Why Vision-Language Models Behave like Bags-of-Words, and What to Do About It?

**Additional Comments On Reviewer Discussion:**

The paper initially got scores of 3, 5 and 8. The authors have provided a rebuttal. The Reviewer VD1i is happy to the rebuttal and kept the score of 8. The Reviewer 1pZP did not attend the rebuttal. The Reviewer qkG7 has partially attended the rebuttal but did not reply the latest response of the authors. The AC has carefully checked the comments of the reviewers and responses of the authors. The AC thinks that all the concerns are solved by the authors.

---

### Decision · Program_Chairs · 2025-01-22

Accept (Poster)